# The Novel Invader *Salpichroa origanifolia* Modifies the Soil Seed Bank of a Mediterranean Mesophile Forest

**DOI:** 10.3390/plants13020226

**Published:** 2024-01-13

**Authors:** Iduna Arduini, Viola Alessandrini

**Affiliations:** Department of Agriculture, Food and Environment, University of Pisa, 56124 Pisa, Italy; viola.alessandrini@phd.unipi.it

**Keywords:** alien plants, invasiveness, lily of the valley vine, resident species, seed bank abundance, species richness

## Abstract

The composition and structure of soil seed banks provide insight into the long-term implications of plant invasions on resident communities. The effect of *Salpichroa origanifolia* (Sa) on the seed bank of a coastal mesophile forest (Tuscany) was studied by growing Sa-rhizomes in soils from low and high invaded sites, in full sun and canopy shade. Sa growth patterns, and the composition, biomass, nitrogen, and phosphorus contents of seedlings which emerged from seed banks were determined. Seed bank abundance and richness were also determined from under and 2 m apart established Sa populations. Sa plants’ leaf traits and biomass allocation changed in response to light conditions. The germination of seed bank seedlings was not affected or even promoted by Sa, while their biomass as well as N and P uptake were more than halved in both light conditions, leading to a progressive depletion of the forest seed bank. Richness was lower under established Sa populations. Sa seedlings exerted a greater suppression on residents than Sa adults, and these appeared more competitive against their own seedlings than on residents. Sa is an invader of concern for Mediterranean forests because of its adaptability to shaded conditions, the competitiveness of its seedlings, and its vegetative spread by means of rhizomes.

## 1. Introduction

Invasive alien species are accounted among the major threats to biodiversity and impose high costs to communities in terms of damage and management [1]. Invasive plants may affect native species through (i) the direct competition for space, light, water, and nutrients, (ii) the release of allelopathic chemicals inhibiting seed germination or seedling growth, and (iii) the alteration in the soil, chemical, and biological environment into less favorable conditions [2]. Among plant traits favoring invasiveness, the most common are great adaptability to variable environmental conditions, high relative growth rate, vegetative reproduction, long flowering season, high propagule pressure, and high variation in seed traits and germination [3,4,5,6]. Nevertheless, plant traits alone cannot fully ensure the success of invasion, which is also related to the characteristics of the invaded community and to environmental factors, so that some ecosystems are more prone to being invaded than others [7]. Experimental evidence did not confirm that overall richness and ecosystem biodiversity are traits of resistant communities, whereas resistance to invasion was associated with the presence of a variety of competitor species [8,9,10]. In all cases, however, resistant communities reduced the abundance of invaders but never avoided their establishment [8]. Moreover, the success of invasion is also influenced by wild herbivores, which were found to contain the spread of invaders, but also to favor them by dispersing propagules, and by selective browsing [8,11].

The production of a persistent seed bank is a primary determinant of the successful establishment of the invader species in the new environment [12]. Therefore, the composition and structure of seed banks following plant invasions can provide valuable insight into the long-term implications of invasion on the resident community. Moreover, the potential vegetation stored in soil seed bank gives a good estimate on the community ability to restore pre-invasion conditions [13,14], which is a crucial information for planning restoration measures. Soil seed banks have a complex structure and function, as they contain seeds ready to germinate, seeds that may need more years or specific events to lose dormancy, and even seeds, mostly arrived from external vegetation, that will not germinate at all because current environmental conditions do not match their requirements. The last mechanism is defined as *storage effect* and allows for species to coexist, taking advantage of spatially and temporally variable environments [15]. This mechanism may be strategic for both alien and native species: the former for colonizing new habitats, and the latter for coping with the environmental changes introduced by aliens [16]. In general, two cohorts of seeds are distinguished: a persistent seed bank to which seeds deposited or produced at the site more than a year earlier belong to, and a transient seed bank formed by the seeds shed in the current and previous year [17,18]. Seeds of a given species may be distributed in both cohorts, with those in the persistent one acting as a propagule reservoir preventing population from extinction [19]. Changes in the abundance and composition of seed banks have often been reported in response to an alien invader, while differences in richness are more difficult to assess because of the spatio–temporal fluctuations in the presence of many species in the seed bank [20].

The recent massive spread of *Salpichroa origanifolia* (Lam.) Baillon (1988) in the Nature Reserve of Migliarino, San Rossore, Massaciuccoli (Pisa, Italy) has given us the opportunity to assess the effects of an alien plant on the soil seed bank of resident species. The forest presents high biodiversity in both the tree and understory vegetation because it extends on an ancient dune system and, therefore, consists of alternating small soil elevations and depressions that create a variety of microclimates at a close distance [21,22,23]. A detailed floristic inventory of the sampling area was carried out by [24] under the toponym ‘Bosco fosso Cuccia’, while the vegetation was described by Tomei et al. [22]. Today, the forest structure is typical of old-growth forests, with vertical and horizontal heterogeneity, large trees, standing dead trees, coarse woody debris on the forest floor, and progressively increasing gaps in the forest canopy [25]. These conditions, along with the high pressure from wild ungulates disturbing the forest floor and dramatically reducing forest recruitment, may favor the arrival and establishment of species from the neighbor anthropized vegetation of agricultural and urban land [13,23]. *S. origanifolia* is native to the temperate regions of south America, present in Bolivia, Peru, Argentina up to North Patagonia, Chile, Brazil, Paraguay, and Uruguay [26]. Even in its native range it displays great environmental adaptability, growing from 0 to 2500 m asl, and is accounted as a noxious weed, frequently occurring in disturbed habitats, at roadsides, and along fences [27,28]. Because of intentional introduction for ornamental purposes, *S. origanifolia* is present in several countries of all continents [28], and in Australia, Japan, and California it is recognized as an invasive plant which could disrupt critical habitats and impact threatened or endangered native species [29,30,31]. Up to today, *S. origanifolia* is not listed as an invasive alien species of EU concern, and it is also not included in any of the EPPO lists [32,33]. However, the spot invasions signaled in Southern Switzerland, on the Canary Islands and in Italy, suggest that its inclusion in both lists should be seriously taken into consideration [34,35,36]. The diffusion of *S. origanifolia* in Tuscany is probably to impute to plants escaped from botanical gardens, and the oldest records in nature date back to 1920 around Florence (*Herbarium Passerini*), and to 1923 along the rocky coasts of Livorno [37]. In the nature reserve, it was first reported by Orlandi and Arduini [38] who found one isolated plant in a recently cut *Pinus pinea* plantation and another one around the entrance buildings. In recent years, *S. origanifolia* has been spread in several parts of the nature reserve, forming dense mono- and pauci-specific layers both within and outside forests.

The present study aimed at investigating the adaptability of *S. origanifolia* (Sa) plants sprouted from rhizomes (Sa adults, hereafter) to full sun and shaded conditions, and to assess the effect of this species on the soil seed bank of the invaded mesophile forest, in the short and long term. We hypothesize that the spread of Sa will lead to a progressive depletion of the soil seed bank of resident species in the invaded forest sites. The short-term effect was investigated by adding sprouting rhizomes to soils collected in Sa-free points of a low and a high Sa-invaded part of the research area. These two sites were chosen to reveal whether the different degrees of Sa-invasion could be related to differences in soil properties or seed bank composition. On the seedlings that emerged from the seed banks, we determined the rate of emergence, the abundance, richness, biomass, and nitrogen and phosphorus concentrations, and contents, after 11 weeks. Changes in seed bank composition in response to treatments were estimated by the relative abundance of growth forms (trees, climbers, dicotyledon herbs, and grasses). The long-term effect of Sa invasion on the resident species was investigated by comparing the abundance and richness of the soil seed banks collected under established Sa populations in different forest sites with those of seed banks collected in Sa-free points at a close distance to each population.

## 2. Results

### 2.1. Short-Term-Effect Experiment

#### 2.1.1. *Salpichroa origanifolia* Adult Plants

The Sa rhizomes buried at the start of the experiment had an average dry weight of 5.8 ± 1.9 g pot^−1^ (means ± SE). The N concentration was 12.3 ± 1.7 mg N g^−1^ and the P concentration 1.6 ± 0.21 mg P g^−1^. Accordingly, the estimated N and P contents were 71.3 ± 8.2 and 9.3 ± 2.6 mg pot^−1^, respectively. After 11 weeks, analysis of variance showed that the interaction Site × Illumination was not significant for any of the variables determined on Sa adults, while significant Site and Illumination mean effects were detected for several variables.

The total dry biomass of Sa adults was 25% higher in HD than LD (Table 1), without differences in the partitioning of biomass between hypogeal organs (rhizomes + adventitious rootlets, hereafter named rhizomes), stems, and leaves, which accounted for, respectively, 63%, 17%, and 20% of the plant biomass. Moreover, no differences between sites were found in leaf traits (chlorophyll concentration, leaf area, and SLA) and these data were not reported. The average N concentration of the entire plant was markedly higher in HD (Table 1) due to the significant differences in N concentration of rhizomes (15.7 in HD and 12.8 mg g^−1^ in LD). The P concentration of the entire plant did not differ between sites, though that of the leaves was significantly higher in HD, 1.6 mg g^−1^, compared to 1.3 mg g^−1^ in LD. Because of the slightly higher biomass, as well as N and P concentrations, the nutrient uptake of Sa plants was 54% and 67% higher for N and P, respectively, in the HD site compared to LD (Table 1).

Illumination affected the biomass of Sa adults, which was 28% higher under canopy shade than in full sun (Table 1). Shade did not change the biomass of rhizomes but increased that of aerial organs: stems changed from 3.8 to 5.4 g pot^−1^ (+42%), and leaves by 60% (Table 2). Accordingly, the partitioning of biomass within plant parts changed in response to illumination and the leaf mass fraction increased while that of rhizomes decreased under canopy shade (Figure 1). We did not count the number of leaves per pot, but we found that the higher leaf biomass of Sa plants grown in shade was associated with a higher surface and weight per leaf, without changes in the specific leaf area (SLA) (Table 2). Leaves of shaded plants also displayed a higher chlorophyll content both on unit surface and unit weight. Finally, the imposed light conditions affected the reproductive biology of Sa, as flowers were observed only in the FS treatment.

Canopy shade also increased the average N concentration of the entire plant (Table 1), primarily because of the markedly higher N concentration of rhizomes, which was 16.2 mg g^−1^ in CS and 12.4 mg g^−1^ in FS. Leaf N concentration was only slightly higher in CS, at 17.5 mg g^−1^ compared to 16.6 mg g^−1^ in FS, while that of stems was not affected and measured 11.9 mg g^−1^, averaged over illumination treatments. The improved biomass production coupled to the higher N concentration led to a 54% higher total N content (Table 1). The total P content was 33% higher in CS than FS (Table 1), which was only due to the higher biomass, as the P concentration was not affected and was 1.6 mg g^−1^ in rhizomes, 1.8 mg g^−1^ in stems, 1.5 mg g^−1^ in leaves, and 1.6 in the entire plant.

#### 2.1.2. Seed Bank Seedlings

##### Richness and Composition

Most seedlings did not reach flowering within the research period, and it was, therefore, difficult to identify all species. Moreover, seedlings belonging to the families *Poaceae* and to the genus *Carex* were grouped into two types only, which probably led to underestimation of the real seed bank richness. Overall, 19 species/types were identified, 10 out of which were common to the two sites, with *Caryophyllaceae* the most represented family (Table 3).

The two sites did not differ appreciably in total richness, with 14 species recorded in LD and 15 in HD, while they differed in the composition in climber and tree species: *Clematis vitalba*, *Rubus* sp., and *S. origanifolia* were detected in both sites, whereas *Periploca graeca* was found only in LD and *Hedera helix* in HD. Concerning trees, seedlings of *Fraxinus* emerged only from the soil seed bank of LD, whereas *Laurus nobilis* and *Ulmus* sp. seedlings only from HD.

The analysis of variance did not evidence significant differences in the number of species per pot in response to treatments, which was, on average, 7.8. Differences were, indeed, detected in the relative abundance of growth forms between sites, and in response to the Illumination × Sa presence interaction. On the LD site, dicot herbs and grasses shared almost equally more than 90% of seed bank seedlings, whereas in the HD site more than half the seedlings that belonged to climbers and dicot herbs did not reach 9% (Table 3). This result did only in part rely on the greater abundance of Sa seedlings, which accounted for 12% of total seedlings in HD and for less than 1% in LD. The relative abundance of grasses and trees did not differ between sites and as expected, the category of trees was by far the least abundant.

##### Seedling Emergence

All treatments affected the emergence of seedlings from the soil seed bank, and ANOVA detected significance for the Site × Time and the Illumination × Sa presence × Time interactions. Seedling emergence was higher in LD than HD up to the fourth week, but then it ceased, while in HD the number of seedlings that emerged continued to increase (Figure 2a).

Accordingly, the number of seedlings was higher in LD from week three to five after start but higher in HD at the end of the experiment, with 72 seedlings per pot compared to 54 in LD. The different emergence in the first weeks could be imputed to the greater proportion of herbaceous species, both dicots and grasses, in LD (Table 3). A greater number of seedlings that emerged in HD after 11 weeks, which was only in part due to Sa seedlings, as they were approximately 9 per pot compared to 0.5 per pot in LD, without any difference in response to illumination and Sa presence. The number of seedlings that emerged was highest in pots exposed to full sun that also contained Sa adults (FS +Sa) and lowest in those exposed to full sun that did not contain them (FS −Sa) (Figure 2b). Intermediate values were recorded under canopy shade, without any difference in the number of seedlings that emerged in response to Sa presence. The differences were significant only for the FS +Sa treatment, with more seedlings per pot recorded after four weeks and from week eight to the end of the experiment.

##### Seedling Abundance

After 11 weeks, the numbers of total and resident seedlings that emerged from the soil seed banks and the relative abundance of growth forms differed significantly in response to the interaction Illumination × Sa presence. The numbers of total and resident seedlings, but not the total and relative abundance of growth forms, also changed in response to the Site × Sa presence interaction. Finally, the number of Sa seedlings was only affected by the site mean effect.

Under canopy shade, the number of resident seedlings was approximately 53 per pot without differences between the −Sa and +Sa treatments (Table 4). In the full sun conditions, conversely, their number was slightly lower than in CS in FS −Sa and increased by 79% in the presence of Sa adults. The number of Sa seedlings was not affected by illumination nor Sa presence, at 4.6 per pot and, therefore, the number of total seed bank seedlings followed patterns of residents being highest in FS +Sa.

The combination of Illumination × Sa presence modified the number of seedlings per growth form and their relative abundance. The numbers of dicot-herbs and grasses were significantly higher in the FS +Sa treatment, whereas the number of tree and climber seedlings was only slightly higher in the presence of Sa adults, and climber emergence was also favored by shaded conditions (Figure 3a). These different patterns changed the relative abundance of growth forms in the seed bank community, with the greatest differences recorded for climbers and dicot herbs, of which relative abundance increased (climbers) and decreased (dicot herbs) significantly in the CS +Sa treatment (Figure 3b). The patterns displayed by the climbers were not related to those of the Sa seedlings, of which the proportion within the climbers ranged from 16% (FS −Sa) to 24% (FS +Sa), without differences in response to illumination and Sa presence. The relative abundance of grasses was 39% across treatments, with only a slight decrease in the presence of Sa under canopy shade (CS +Sa) (Figure 3b).

The numbers of resident and total seedlings that emerged from the seed bank were higher in the HD +Sa treatment compared to the others, whereas the number of Sa seedlings differed only in response to site, being 0.5 in LD and 8.8 seedlings pot^−1^ in HD (Figure 4a). Resident seedlings represented 99% of the total seed bank in LD, and 88% in HD, without differences due to the presence of Sa adults.

##### Seedling Biomass

Oppositely to the effect on germination, the presence of Sa adults greatly reduced the cumulative biomass of seed bank seedlings, which was 17.5 g pot^−1^ in −Sa and only 7.7 g pot^−1^ in +Sa, whiteout any significant interaction with light and site treatments. Differences in response to site and illumination were found for the biomass of resident and Sa seedlings, which changed their proportion in the seed bank biomass.

The presence of Sa adults decreased the biomass of residents only under canopy shade, while that of Sa seedlings in both light conditions (Table 4). In consequence, resident seedlings accounted for 82% of total seed bank biomass in FS +Sa, while their proportion decreased to 67% in CS +Sa and was only approximately 56% in both −Sa treatments (Table 4).

A significant interaction Site × Sa presence was found for the biomass of resident and Sa seedlings. In the LD site, the biomass of residents was more than halved in the presence of Sa adults, without any influence of Sa seedlings, which biomass was only 6 mg pot^−1^ in +Sa and 0 in −Sa (Figure 4b). In the HD site, the biomass of resident seedlings was always lower than in LD, and it is remarkable that the lowest biomass, 2.9 g pot^−1^, was achieved in the −Sa treatment. The biomass of Sa seedlings showed opposite trends, being fourfold greater in absence of Sa adults (Figure 4b). Therefore, in HD the proportion of residents in the seed bank biomass was 55% in the presence of Sa adults and only 16% in their absence, which highlights that Sa seedlings reduced the growth of resident seedlings more than Sa adults.

##### N and P Concentrations and Content of Seed Bank Seedlings

Illumination affected the N concentration of resident seedlings, which was significantly higher in shaded conditions, 15.7 mg g^−1^, than in full sun, 11.5 mg g^−1^, without interaction with the other treatments. The N concentration of Sa seedlings was not affected by light conditions, and was, on average, 12 mg g^−1^. A significant interaction Site × Sa presence was detected for the N concentration of resident and Sa seedlings (Figure 5). In LD, the N concentration of residents was slightly lower in +Sa, while the opposite occurred in HD, so that the N concentration was significantly higher in HD +Sa (15.2 mg N g^−1^) than in LD +Sa (11.5 mg N g^−1^). The N concentration of Sa seedlings was markedly lower in +Sa than in −Sa, 7.4 compared to 11.9 mg N g^−1^, suggesting the existence of intraspecific competition for N uptake between Sa adults and seedlings. The P concentration of residents differed only in response to site, being 3.2 mg g^−1^ in LD and 2.6 mg g^−1^ in HD; that of Sa seedlings was slightly lower, 2.2 mg g^−1^, and did not differ among treatments.

The total N and P contents of seed bank seedlings changed in response to the interaction Illumination × Sa presence. Under canopy shade, the N and P contents of residents were markedly reduced by the presence of Sa adults, while in full sun they did not differ between +Sa and −Sa treatments, and values were intermediate those recorded in CS for N, and similar those of CS −Sa for P (Table 4). As Sa seedlings changed only in response to Sa presence, the total N and P contents of total seed bank seedlings was lower in the presence of Sa adults, but the differences between the +Sa and −Sa treatments were much more pronounced under canopy shade than in full sun (Table 4). The partitioning of the total N and P accumulated by seed bank seedlings between resident and Sa seedlings was also affected. In the presence of Sa adults, residents contained 87% and 75% of total N and P in FS and CS, respectively, whereas these proportion decreased markedly in −Sa, especially for N (Table 4).

The presence of Sa adults reduced the total N content of seed bank seedlings from 231 to 92 mg pot^−1^, and the total P content from 47 to 22 mg pot^−1^, without differences between sites. Due to the higher biomass of Sa seedlings in HD, a significant interaction Site × Sa presence was found for the N and P contents of residents, and for both elements, the highest contents were recorded in the absence of Sa adults and also Sa seedlings (LD-Sa), and the lowest when only Sa seedlings were present (HD-Sa) (Figure 4c,d). Oppositely, the N and P contents of Sa seedlings were sixfold (N), and fourfold (P) lower in +Sa than in −Sa. As a result, in the HD site, resident seedlings contained approximately the 64% of the total N and P accumulated by the seed bank in the presence of Sa adults, but only 17% in their absence.

### 2.2. Long-Term-Effect Experiment

Over the entire experiment, we identified 27 different species, 10 out of which were found in the soils collected at a 2 m distance (control) from Sa populations but not under them (Sa-covered) (Table 5). The number of species recorded over all sampling points was markedly lower under Sa populations in all germination tests (Table 5). Values were lower in April than in June, probably due to the lower number of sampled points, which were five and ten, respectively. Cold pre-treatment almost doubled the number of species that emerged from the soils collected in June, both from the control and Sa-covered seed banks. In contrast to the short-term-effect experiment, Sa seedlings were never recorded.

The number of seedlings germinated at laboratory conditions was always lower in the Sa-covered soils, with significant differences to controls in the April and June tests (Figure 6a–c). In the Sa-covered treatment, the average number of seedlings per tray was only 12% and 28% of that recorded in the control soil, respectively, in April and June (Figure 6a, b). After cold pre-treatment, the emergence from Sa-covered was 79% of that of the control (Figure 6c). A similar result was observed for richness: soils collected under Sa populations always exhibited a reduced average number of species compared to soils collected adjacently, with significant differences in the soils collected in April and in June after cold pre-treatment (Figure 6d–f).

The presence of Sa populations reduced both dicotyledons and grasses and, because of the sporadic presence of most species, our results could not demonstrate differential selection of Sa against given species. However, it is worth noting that no grasses emerged from the seed bank of Sa-covered soil in April (Figure 6a), and that their average proportion among seedlings was very low in the cold pre-treated June soil (Figure 6c), thus suggesting a greater sensitivity to Sa invasion.

## 3. Discussion

Phenotypic plasticity, fast growth, and allelopathy are among the most cited traits driving the success of invasive species over natives [2,39,40,41]. Moreover, there is increasing alert that plasticity in response to light conditions may also support aliens in colonizing undisturbed understory vegetation [42,43]. According to the ecological indicator values assessed by Landolt [34], *Salpichroa origanifolia* (Sa) is classified as a semi-shade (L3) species for light reaction. In our research, Sa plants adapted well to both full sun and shaded conditions, changing leaf morphology and the partitioning of biomass between leaves and rhizomes. They also changed reproductive biology, in that flowering was observed only under elevated illumination in the short-term-effect experiment and also in forest clearings. Plants grown in full sun allocated proportionally less biomass into leaves by reducing leaf size, while the specific leaf area did not change in response to light conditions. The values of SLA recorded in Sa adults (21 mm^2^ mg^−1^) were higher than those generally reported for natives and are typical of exotic invasive species with a greater potential for fast growth [44]. This plasticity in response to light may be a key trait explaining the great ability of Sa to colonize different habitats [28,34]. The canopy-shade conditions reduced on average up to 90% in full sun illuminance, which corresponded to the light-reductions reported for many closed-canopy forests [42,45]. The good performance of both Sa adults and Sa seedlings in low light conditions complies with the vigorous growth observed in the invaded forest and indicate that this species may be able to successfully colonize closed-canopy forests, in contrast to the belief that undisturbed ecosystems are resistant to alien invasion [42,45].

In addition to light, other abiotic factors, such as temperature, soil moisture, and nutrient availability are the most important barriers filtering the colonization of alien species within new habitats [39]. Located in the typical Mediterranean climate region, the mesophile forest of San Bartolomeo did not present extreme conditions preventing Sa colonization, and the soils collected in the sites with a low (LD) and high (HD) density of Sa plants showed a similar pH and C/N ratio, as well as slightly differing in soil texture, organic C, total N concentration, and cation exchange capacity. Our results highlight, however, that the biomass and the N and P content of Sa adults and, especially, Sa seedlings were higher in HD at the end of the short-term-effect experiment. Moreover, in all examined plant cohorts, the N concentration was slightly higher in HD than LD in Sa adults, Sa seedlings, and resident seed bank seedlings, despite the total N concentration in soil being lower in the former site. These results suggest that more complex soil–plant interactions favored both N and P uptake in HD and greatly enhanced the growth of Sa seedlings. Gaining deeper insight into the response of Sa adults and seedlings to different soil types would be useful to assess to what extent physical and chemical soil properties may contribute to the spread of Sa in natural environments, and to identify those in which the species may become invasive. While the more abundant emergence of Sa seedlings from HD soils complies with the more diffused presence of the invader species, their vigorous growth compared to LD suggests either that the soil properties are more favorable for Sa growth or that the resident community is less competitive [39]. According to the ecological indicator values, Sa prefers dry soils (F 1.5) with small variations in soil moisture (W 1) [34], specifically, conditions that better realize in the soil with higher sand content (HD).

Resident communities are rarely able to prevent invader establishment, while they may constrain the spread of alien populations [39]. However, which traits make resident communities resistant to invasion is still under debate, and the effect of invasion on soil seed banks differs with the identity of the invading species and with the characteristics of invaded communities [8,9,20]. The classic niche theory predicts that the establishment of an alien species is favored the more it differs from natives in the use of resources, while it is hindered by similar competitors [46]. On the other hand, similarity to non-native resident species, it was found to increase the success of new invaders [47], and both human and wildlife disturbance may favor alien invasion [11]. Our study could not demonstrate differential resistance to Sa invasion in the LD and HD seed bank communities. Nevertheless, we found that the community with a higher Sa density (HD) was more than half composed by seeds of climber species, inducing us to hypothesize that the overall abiotic and biotic conditions were particularly favorable to this growth form, thus favoring Sa stabilization. This point needs to be disentangled as it may depend on several factors, but in a semi-arid sub-tropical continental forest, the composition and distribution of climber species was found to be associated with soil texture and moisture [48].

Differing from what was hypothesized for many invasive species [2,49], a strong allelopathic effect could be excluded both for Sa adults and seedlings, as we did not find any differences in seedling emergence in shaded conditions, and germination even seemed to be stimulated by the presence of Sa adults in full sun. Similar to Sa, the invasive *Oxalis pes-caprae* promoted the emergence of native seedlings, which was imputed to facilitative effects on P availability [50]. We think it is more likely that in full sun, the presence of Sa promoted germination of forest floor species by reducing daily fluctuations in soil temperature and moisture, thus maintaining a more favorable light and temperature environment [43]. Invasion driven by allelopathy was also excluded for the invasive grass *Cenchrus echinatus*, in which the formation of dense covers reduced the germination of resident seeds [41]. However, as Sa leaves produce withanolides with known antifungal and insecticidal properties, targeted investigations are needed to unravel whether the release of these compounds could exert a direct or indirect selective effect on given plant species [51,52,53].

While the germination of seed bank seedlings was not affected or even promoted by Sa, their growth was greatly reduced and, like in *Cenchrus echinatus* [41], we identified a high growth rate as the primary driver of competition of Sa over resident species. At the end of the 11-week experiment, Sa plants which sprouted from rhizomes had produced dense covers, in which leaf biomass alone equaled the cumulative biomass of resident species in low light conditions. In full sun, the competitive ability of Sa was slightly reduced, which was in part because of the positive effect of Sa on the emergence of seed bank seedlings and because of the smaller leaves. While a high growth rate is expected and well known for rhizomatous plants [54], Sa was demonstrated to be very competitive even at the seedling stage, and Sa seedlings accounted for 72% of the total seed bank biomass, despite representing only 12% of total seedlings.

Unexpectedly, the competitiveness of Sa seedlings towards the resident community was greatly reduced by Sa adults that seemed to exert a greater inhibition against their own seedlings than on residents. Indeed, the biomass of Sa seedlings that emerged in the presence of Sa adults was only 23% of that obtained in their absence and, as the N concentration also decreased markedly, we hypothesize that intraspecific competition was primarily for N uptake. The present study could not fully disentangle this point because of the interference with resident species, in which biomass production in the presence of only Sa adults was not available for the HD site, and comparisons with LD are merely indicative because of the differences in composition.

In our experiment we did not follow the fate of seed bank seedlings as we harvested them before most species reached flowering. However, the huge biomass reduction allows for inference that the allocation of resources into reproduction would be greatly impaired especially in shaded conditions [50]. Thus, while the dense covers formed by *Cenchrus echinatus* reduced the germination of resident seedlings maintaining them in the seed bank [41], and *Oxalis pes-caprae* did not alter seed rain-maintaining seed bank abundance and richness over years [50]. In our research the number of seedlings and species that emerged was systematically lower in soils collected under established Sa populations, suggesting a progressive impoverishment of the seed bank. Moreover, we found a markedly lower emergence of grasses from Sa-covered soils in two out of three germination tests (April and June 5 °C), which could reveal a higher sensitivity of this functional group compared to dicot herbs. Moreover, though a selective sensitivity of given species to Sa invasion could not be demonstrated in our study, a shift in composition towards exotic/invasive seeds could be expected [55], similar to that observed in the same area in response to disturbance [13].

Propagule pressure associated with the production of a persistent seed bank is widely recognized as a key factor of invasion success, allowing for colonization and later the establishment of stable populations [10,12]. The germination ecology of Sa has not been investigated extensively by the literature, but according to Galetto [56] the germinability is low, approximately 11% at 22 °C and in a 12 h photoperiod. Considering that flowering and fruiting occur only in elevated light conditions, the contribution of seedling number to propagule pressure is expected to be low. It must be underlined, however, that Sa emergence was recorded in soils collected just before the outbreak of spring in the understory community (5 March), but not later (23 April and 19 June). This suggests that the germination time of Sa coincided with that of most species of the mesophile forest, which may further enhance its competition for resources with the resident community at initial growth stages. The average N and P concentrations of resident seedlings was generally higher than that of Sa adults and seedlings, especially in shaded conditions. Despite this, the amounts of N and P accumulated by Sa seedlings were much higher, and nutrient shortage probably also contributed to limiting the growth of residents. In summary, the propagule pressure of Sa can be considered low in terms of the number of seedlings, but very high in terms of seedling competitiveness for space and resources, as the high growth rate of seedlings at initial stages is closely followed by vigorous vegetative growth by means of rhizomes, allowing for Sa plants to spread and rapidly cover the forest floor. Therefore, great attention should be paid to identifying transport pathways, so as to reduce the arrival of seeds and rhizome fragments associated with human activities [9].

## 4. Materials and Methods

### 4.1. Research Area

The research area, named ‘Forest of San Bartolomeo’ is a special protected area extending for approximately 120 ha at the eastern border of the ‘Forest of San Rossore’, which is in the central part of the 24,000-hectare Nature Reserve of Migliarino, San Rossore, Massaciuccoli (Figure 7). The reserve is situated along the coastal plain of Tuscany (Italy), at coordinates 43°43′ N and 10°16′ E. Since 2004, it has been included in the Biosphere Reserve ‘Selve Costiere di Toscana’ due to its rich biodiversity and the demonstrated harmonious relationship between humans and nature. The nature reserve extends on ancient dunes running parallel to the coastline, interspersed with small lagoons or bogs, and the soil consists of unconsolidated marine sediments and alluvial deposits of the rivers Arno and Serchio. According to the Köppen classification, the climate is characterized as a hot and humid Mediterranean climate (Csa). The average mean annual maximum and minimum daily air temperatures are 20.2 °C and 9.5 °C, respectively, with an annual precipitation of 971 mm mainly concentrated in the autumn and winter months [57]. The photoperiod varies from 8 h and 46 min to 15 h and 14 min. The nature reserve comprises a diverse range of ecosystems, including conifer plantations (*Pinus pinea* L. and *Pinus pinaster* Ait.), mixed broadleaved forests dominated by *Quercus ilex* L., *Q. robur* L., and *Fraxinus angustifolia* Vahl, extensive forest clearings, wetlands, and agricultural land [22].

Once, the forest of San Bartolomeo was fenced to support the natural regeneration of forest trees, while it now undergoes heavy pressure from deer and wild boar. The vegetation belongs to the type *Fraxino angustifoliae-Quercetum roboris* Gellini, Pedrotti, Venanzoni [58], with the subass. *carpinetosum betuli* and the variant with *Pinus pinea* in the dominant layer [22].

In recent years, forest gaps caused by natural treefall were invaded by *Salpichroa origanifolia* (Lam.) Baillon (1988) (Sa), common name ‘lily of the valley vine’. Sa is a perennial herb of the *Solanaceae* family, which produces a network of rhizomes and long aerial branches with prostrate or ascending habit [26]. Leaves are petiolate, with a simple rhomboid lamina that is sparsely covered by glandular trichomes containing the sugar ester acyl sucrose [27] and withanolide compounds involved in the plant defense system against insects, bacteria, and fungal pests [27,51]. In the native range, the species displays profuse and long flowering and fruiting, lasting up to 6–7 months, beginning from spring to autumn [56]. Flowers are white, solitary, bell-shaped, and approximately 1 cm long. Fruits are creamy-white, 1–2 cm long, with ovoid berries, which contain from 10 to 20 seeds, and are dispersed by birds [59].

### 4.2. Experiments

In two independent experiments we assessed: (i) the short-term effects of the addition of sprouting Sa rhizomes on the seed bank of a not invaded soil, and (ii) the changes in the soil seed bank under established populations of Sa compared to not-invaded sites (long term effect). The experiments were carried out in 2023 at the Department of Agronomy and Agroecosystem Management of the University of Pisa (43°41′ N, 10°23′ E, Tuscany, Italy). Photoperiod and climate conditions were similar those described for the sampling area.

The methods for analyzing seed banks can be roughly distinguished in germination and extraction methods. Both have advantages and disadvantages, as germination methods underestimate the number of seeds but reveal more species, whereas extraction methods also count unviable seeds and fail in detecting small seeds [60]. We chose to analyze seed banks by the germination method because it could provide a more complete list of species compared to the extraction methods when seed banks are mostly composed by very small seeds and the soils are rich in organic matter [19,61], which is expected for this forest [13].

#### 4.2.1. Short Term Effect Experiment

##### Experimental Design

To quantify the short-term impact of Sa on the emergence and growth of seedlings from the soil seed bank of the San Bartolomeo Forest, we arranged a fully factorial experimental design with three crossed treatments and three replicates, in which the presence/absence of Sa rhizomes (Sa presence) was tested at two light conditions (Illumination) on the soil seed bank of two sampling sites (Site). The forest soil was collected from the 0–5 cm soil layer before the sprouting of most understory species (5 March 2023). Two sites of the research area, similar in overstory vegetation but differing in invasion, were chosen: one with sporadic Sa plants covering for less than 10% the forest floor (low density site, LD) and the other with established Sa populations covering approximately 40% of the forest floor (high density site, HD). In both sites, the soil was sampled 5 m apart from Sa plants to exclude previous influence of the alien, and 15 samples per site were collected and mixed to overcome spatial heterogeneity of seed distribution. Soil properties referred to in the 0–10 cm profile were determined following standard methods [62] (Table 6).

Once cleansed from gravel and coarse litter debris, soils were spread into 10 L pots (50 cm length, 16 cm width, 16 cm height) which were previously filled with a 10 cm layer of expanded clay that was covered by a 0.9 mm mesh to avoid soil and seed loss. Each pot received 2 L of forest soil distributed to form a layer of 585 cm² with an average depth of 3.4 cm. In total, we arranged 24 pots, 12 with the LD and 12 with the HD soil (Site treatment). To arrange the ‘Sa presence’ treatment, nine 10 cm fragments of Sa rhizomes were buried in six pots from each site treatment (+Sa), while the other six pots were used as controls without Sa rhizomes (−Sa). Rhizomes were collected in the HD site on the same date as soil, and three additional samples of nine fragments each were oven dried to determine the dry weight, the N and P concentrations, and the content of rhizomes at the start of the experiment. All pots were placed outdoors and regularly watered by means of drip irrigation. For the Illumination treatment, three pots of each combination Site × Sa presence (HD +Sa, HD −Sa, LD +Sa, LD −Sa) were placed in full sun (FS) and the other three under canopy shade (CS). The latter condition was simulated by covering the pots with shading nets and positioning them under the canopy of a *Platanus hispanica* Mill. ex Münchh. Illuminance was measured 10 cm above the soil surface using the Luxometer PLXM A1, Parkside (Neckarsulm, Germany). As shown in Table 7, the decrease in illuminance between FS and CS treatments became more pronounced over time, thus mimicking forest floor conditions. The net initially reduced illumination by approximately 50%, and as the plane tree began to leaf out, the reduction increased up to 98%.

##### Measurements

The number of emerged seedlings was recorded at week intervals. After 11 weeks, the Sa plants grown from rhizomes (Sa adults) and the seedlings that emerged from the soil seed banks were gently extracted by watering the soil, counted, identified for species or types, and grouped into the growth forms: trees, climbers, dicotyledon herbs (dicot herbs), and grasses. Species identification and the assignment to growth forms followed Pignatti et al. [63,64,65,66], except for Sa that was classified as a climber Geophyte and not a Chamaephyte, according to the literature of the native range [26] and our own observations. The number of seedlings belonging to each growth form was determined, and the relative abundance was calculated as the percentage of seedlings assigned to a growth form on total seedlings. Seed bank richness was estimated as the number of species and/or types per pot.

Just before plant harvest, the chlorophyll concentration of Sa adults was measured by means of the At LEAF CHL BLUE device (FT Green LLC, Wilmington, NC, USA) on 30 representative leaves per pot. Leaves were detached, scanned, and the leaf area was measured using the free software ImageJ (version 1.53a http://imagej.nih.gov/ij, Java 1.8.0_172, accessed on 22 May 2023). Their dry weight was determined, and the specific leaf area (SLA) was calculated as the ratio of surface and weight (mm^2^/mg).

The fresh weight of Sa adults, separated into rhizomes, stems, and leaves, and that of Sa and resident seedlings that emerged in each pot were determined. The leaves sampled for SLA determination were added to the corresponding leaf mass of Sa adults. The dry weight was determined after oven drying all samples at 65 °C to constant weight. The dried samples were milled into fine powder using the Ultra Centrifugal Mill ZM 2000 (Retsch, Haan, Germany) at 16,000 revolutions per minute (rpm) and employing a 0.5 mm Conidur distance-sieve. Powders were mineralized at high temperature with concentrated sulphuric acid, and the nitrogen and phosphorus concentrations were determined by the modified Kjeldahl method and the ammonium–molybdophosphoric blue color method, respectively. The N and P contents were computed by multiplying the N or P concentrations by the respective dry matters.

#### 4.2.2. Long Term Effect Experiment

##### Soil Collection

To evaluate seed bank depletion in the forest floor covered by Sa in the San Bartolomeo Forest, we collected soil samples under established Sa populations (Sa-covered) and at approximately 2 m from each (control). We considered that populations were established when they formed a continuous cover of at least 1 square meter. The soil was collected at five forest sites on 23 April, and at ten sites on 19 June. At each site, three Sa-covered and three control soil samples were collected from the 0–5 cm soil layer, cleaned from gravel and coarse litter debris, mixed, and placed to germinate just after. To improve the germination of spring-dispersed seeds, subsamples of the soils collected in June were cold pre-treated at 5 °C for three months.

##### Germination Tests

Germination tests were carried out in 1 L trays (17.2 cm × 12.5 cm × 5 cm) filled with 400 mL expanded clay covered by a 0.9 mm mesh, on which 250 mL of forest soil was spread in a thin layer. Trays were placed in the laboratory, regularly watered, exposed to room temperature, and illuminated with fluorescent light (Green 58/77, Osram, Milan, Italy) for 12 h per day in the April test, and for 16 h per day in the June tests (Figure 8). Illumination was set to mimic the light conditions under the forest canopy in spring–summer and was approximately 3500–4500 lx. The room temperature ranged from 23 to 28 °C during the experiments.

##### Measurements

After four weeks, the number of seedlings per tray was recorded, and seedlings were identified as species or types. Species/types were grouped into Dicotyledons and Grasses, the former including the categories trees, climbers, and herbs, the latter including the families *Poaceae*, *Cyperaceae*, and *Juncaceae*. At the end of each germination test, species not identified were left to develop further. During this period, we did not observe the emergence of new seedlings.

### 4.3. Statistical Analyses

Statistical analyses were performed with JMP Pro 17.0.0. (SAS statistical package, Copyright © 2022 JMP Statistical Discovery LLC., Cary, NC, USA) using the fit model function, and all treatments were set as fixed effects [67]. Before analyses, models were evaluated to ensure they met the assumptions of independence and normality of residuals and, when necessary, response variables were transformed. In the short-term-effect experiment, data were analyzed by means of ANOVA. To test results on Sa adults, data were arranged in a split-plot design with ‘Site’ as the main plot and ‘Illumination’ as subplot, with three replicates. To test results on seed bank seedlings, data were arranged in a split-split-plot design with ‘Site’ as the main plot, ‘Illumination’ as subplot, and ‘Sa presence’ as sub-subplot with three replicates. To analyze the number of seedlings that emerged weekly during the germination test, the effect ‘Time’ was added as sub-sub-subplot. The post hoc Tukey test was used for comparison among treatments and significantly different means were separated at the 0.05 probability level [67]. In the long-term-effect experiment, data were analyzed by means of the t-test separately for each germination test (April, June, and June 5 °C), with the sampling sites used as replicates.

## 5. Conclusions

In conclusion, the results of the short-term-effect experiment demonstrated that *S. origanifolia* did not affect the germination of resident species, whereas it strongly decreased their biomass. Thus, as hypothesized, a progressive depletion of the soil seed bank is to be expected in consequence of Sa diffusion in the understory of mesophile forests. Though the invasion is still recent, the results of the long-term-effect experiment confirmed an impoverishment in the size and richness of the soil seed bank in the invaded forest sites, with a lower number of seedlings belonging to less species emerging from the soils collected under established populations of *S. origanifolia*. Our results also highlighted that the interaction between invaders and residents is complex, being influenced by light conditions and by both interspecific and intraspecific competition. We found, indeed, that Sa seedlings exerted a more adverse effect on resident seedlings compared to Sa adults, and a strong intraspecific competition occurred between Sa adults and seedlings, which apparently mitigated the negative effect on residents.

As the effects on the biomass of understory species were similar in the two sites of the mesophile forest located along the Tuscan coast (Central Italy), despite the differences in floristic composition, our results demonstrate that *S. origanifolia* is an invader of great concern for the Mediterranean region, which could seriously damage forest ecosystems thanks to its fast growth rate at the seedling stage and its high phenotypic plasticity, allowing for it to adapt to both full sun and shade conditions. As we found that flowering was limited to high-light conditions, control measures should primarily address the removal of aerial branches in illuminated sites during the flowering period, so as to avoid further spreading by means of zoochores’ fruit dispersal.

## Figures and Tables

**Figure 1 plants-13-00226-f001:**
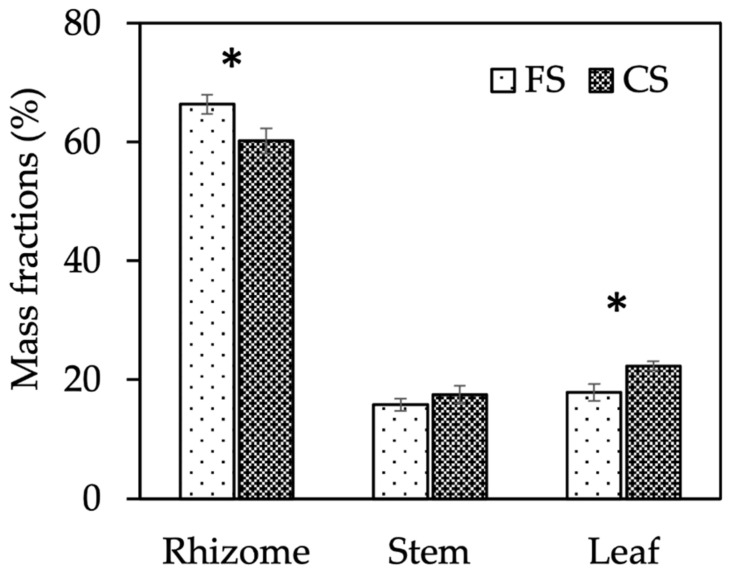
Short-term-effect experiment: mass fractions of rhizomes, stems, and leaves of *S. origanifolia* adult plants as affected by the Illumination treatment mean effect. Data are means ± SE of two sites and three replicates (*n* = 6). FS, full sun; CS, canopy shade. *, indicates significant difference for the corresponding mass fraction at *p* ≤ 0.05, Tukey test.

**Figure 2 plants-13-00226-f002:**
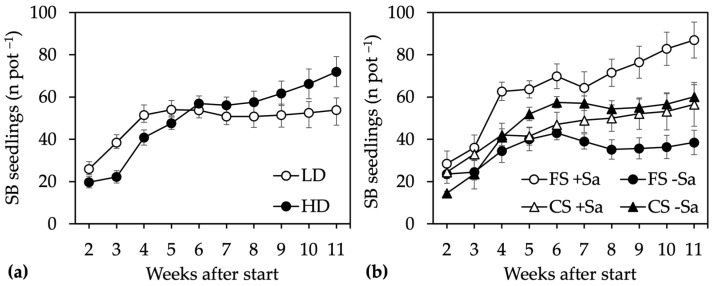
Short-term-effect experiment. Number of seedlings that emerged from soil seed banks (SB): (**a**) as affected by the Site × Time interaction, with data means ± SE of two light conditions, two Sa presence treatments, and three replicates (*n* = 12); (**b**) as affected by the Site × Illumination × Time interaction, with data means ± SE of two sites and three replicates (*n* = 6). LD, low density site; HD, high density site; FS, full sun; CS, canopy shade; +Sa, presence of Sa adults; −Sa, absence of Sa adults.

**Figure 3 plants-13-00226-f003:**
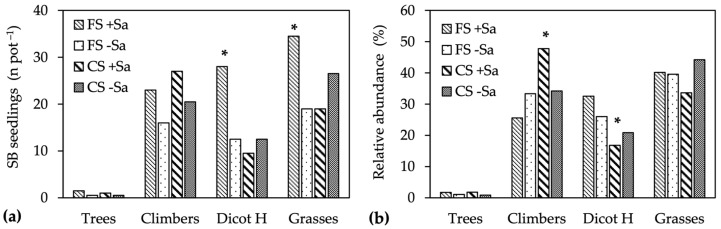
Short-term-effect experiment: (**a**) number of seedlings for each growth form, and (**b**) relative abundance of growth forms in the seed bank communities. Data area means of two sites and three replicates (*n* = 6). Within a growth form, the * superscript bar differs significantly at *p* ≤ 0.05, Tukey test.

**Figure 4 plants-13-00226-f004:**
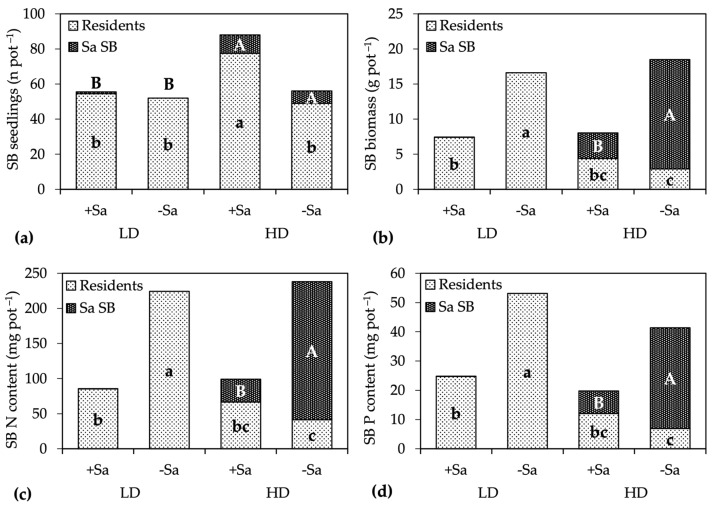
Short-term-effect experiment: (**a**) number, (**b**) biomass, and (**c**) N, and (**d**) P content of seed bank (SB) seedlings partitioned into resident and *S. origanifolia* (Sa SB) seedlings, as affected by the Site × Sa presence interaction. Data area means of two sites and three replicates (*n* = 6). Within a series, bars with the same letter are not significantly different at *p* ≤ 0.05, Tukey test. Lowercase letters indicate significance for resident species; uppercase letters for Sa seedlings. In the Sa SB series, the letters C were not indicated. LD, low density site; HD, high density site; +Sa, presence of Sa adults; −Sa, absence of Sa adults.

**Figure 5 plants-13-00226-f005:**
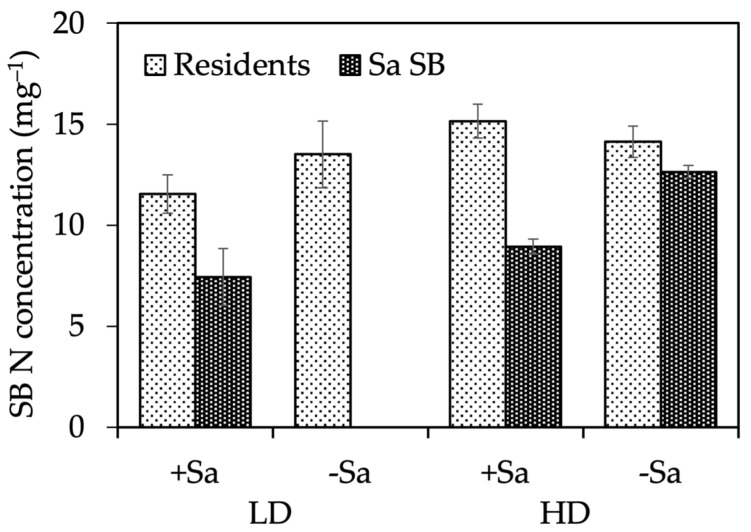
Short-term-effect experiment: nitrogen concentration of resident and *S. origanifolia* (Sa SB) seedlings, as affected by the Site × Sa presence interaction. Data area means ± SE of two light conditions, and three replicates (*n* = 6). LD, low density site; HD, high density site; +Sa, presence of Sa adults; −Sa, absence of Sa adults.

**Figure 6 plants-13-00226-f006:**
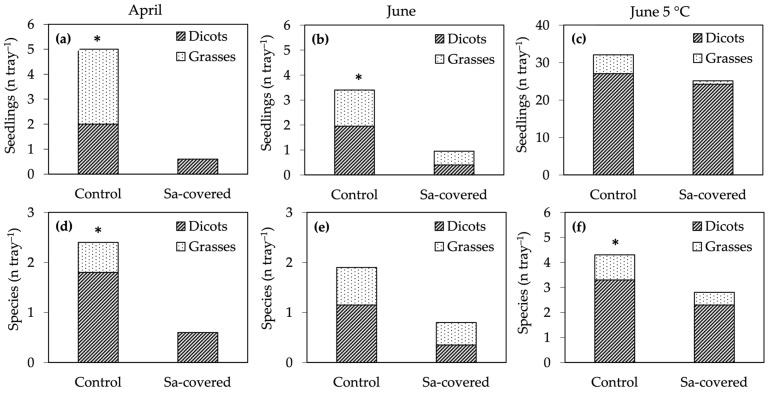
Long-term-effect experiment: (**a**–**c**) number of seed bank seedlings, and (**d**–**f**) number of species that emerged from soils collected under established *S. origanifolia* populations (Sa-covered) and at a 2 m distance (control). Soils were collected in April (**a**,**d**) and in June, tested immediately (**b**,**e**) and after cold pre-treatment (**c**,**f**). Values are means of five replicates (April) and ten replicates (June, June 5 °C). * indicates significant differences for the total SB (Dicots + Grasses) at *p* ≤ 0.05, *t*-test.

**Figure 7 plants-13-00226-f007:**
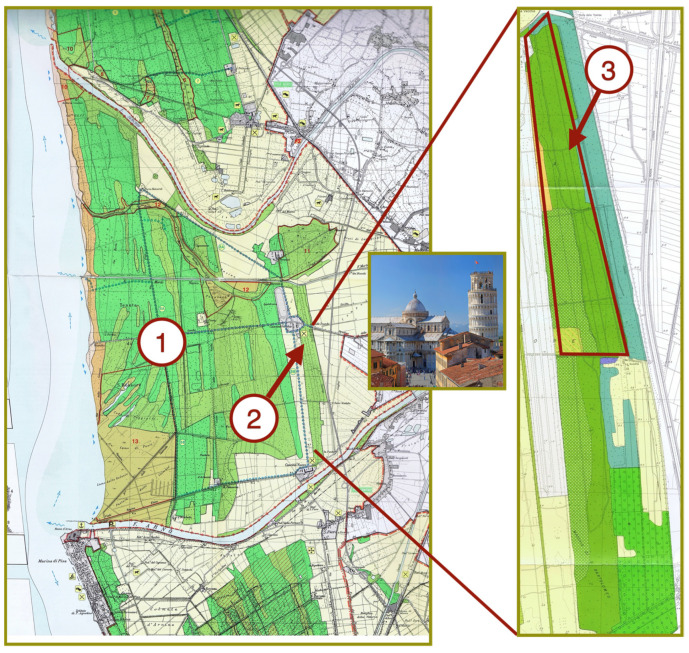
Geographical location of the research area within the Nature Reserve of Migliarino, San Rossore, Massaciuccoli (Pisa, Italy): 1, The Forest of San Rossore; 2, The special protected Forest of San Bartolomeo; 3, The area invaded by *S. origanifolia* where soil samples were collected.

**Figure 8 plants-13-00226-f008:**
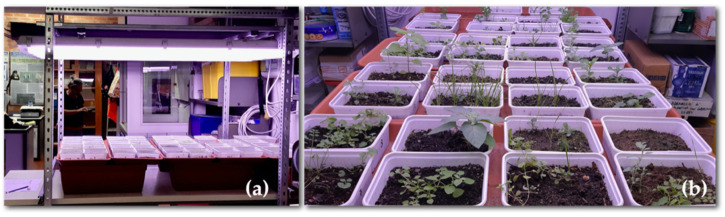
Long-term experiment: (**a**) facility for germination tests in the laboratory; (**b**) the germination test on cold pre-treated soils (June 5 °C) after four weeks.

**Table 1 plants-13-00226-t001:** Short-term-effect experiment: biomass, and N and P concentrations and contents of *S. origanifolia* plants sprouted from rhizomes (Sa adults), as affected by the Site and Illumination mean effects. Within each treatment, values followed by different letters in a column are significantly different at *p* ≤ 0.05, Tukey test (*n* = 6). LD, low density site; HD, high density site; FS, full sun; CS, canopy shade.

Treatment	Sa Adults Biomass	Concentration		Content	
		N	P	N	P
	g dw pot^−1^	mg g^−1^	mg g^−1^	mg pot^−1^	mg pot^−1^
**Site**					
LD	24.7 a	13.3 a	1.5 a	326.7 b	37.7 a
HD	30.7 a	15.6 a	1.6 a	478.8 a	50.0 a
**Illumination**					
FS	24.3 a	13.1 b	1.5 a	317.5 b	37.7 a
CS	31.0 a	15.7 a	1.6 a	488.1 a	50.0 a

**Table 2 plants-13-00226-t002:** Short-term-effect experiment: leaf parameters of *S. origanifolia* adult plants as affected by the Illumination treatment mean effect. Data are means of two sites and three replicates (*n* = 6). Values followed by the same letter within a column are not significantly different at *p* ≤ 0.05, Tukey test. FS, full sun; CS, canopy shade; SLA, Specific Leaf Area.

Illumination	Leaf Dry Weight		Leaf Area	SLA	Chlorophyll Concentration	
	g pot^−1^	mg leaf^−1^	cm^2^ leaf^−1^	mm^2^ mg^−1^	μg cm^−2^	mg g^−1^
FS	4.3 b	14.7 b	3.0 b	20.0 a	10.9 b	0.22 b
CS	6.9 a	22.2 a	4.9 a	21.9 a	18.1 a	0.41 a

**Table 3 plants-13-00226-t003:** Short-term-effect experiment: species recorded at least twice in the seed bank of soils with a low (LD) and high (HD) density of *S. origanifolia*. x, present; -, absent. Relative abundance of growth forms, and of *S. origanifolia*, as affected by the site mean effect. For each growth form, values followed by the same letter are not significantly different at *p* ≤ 0.05, Tukey test (*n* = 12). Soils were collected in the San Bartolomeo Forest on 5 March.

Growth Form/Species	Family	LD	HD
Trees			
*Laurus nobilis*	Lauraceae	-	x
*Fraxinus angustifolia*	Oleaceae	x	-
*Ulmus* sp.	Ulmaceae	-	x
Climbers			
*Clematis vitalba*	Ranunculaceae	x	x
*Hedera helix*	Araliaceae	-	x
*Periploca graeca*	Apocynaceae	x	-
*Rubus* sp.	Rosaceae	x	x
*Salpichroa origanifolia*	Solanaceae	x	x
Dicotyledon herbs			
unknown cfr *Sonchus* sp.	Asteraceae	x	x
*Cerastium* sp.	Caryophyllaceae	x	-
*Moehringia trinervia*	Caryophyllaceae	x	x
*Stellaria media*	Caryophyllaceae	x	x
*Trifolium* sp.	Fabaceae	x	x
*Oxalis corniculata*	Oxalidaceae	-	x
unknown cfr *Veronica* sp.	Plantaginaceae	x	-
*Persicaria hydropiper*	Polygonaceae	x	x
unknown	Solanaceae	-	x
Grasses			
unknown cfr. *Poa annua*	Poaceae	x	x
*Carex* sp.	Cyperaceae	x	x
Relative abundance (% of total seedlings)			
Trees		0.9 a	1.7 a
Climbers		7.9 b	54.2 a
Dicot herbs		46.5 a	8.7 b
Grasses		44.7 a	35.4 a
*Salpichroa origanifolia*		0.9 b	12.2 a

**Table 4 plants-13-00226-t004:** Short-term-effect experiment: number, biomass, and N and P content of resident, *S. origanifolia*, and total seed bank (SB) seedlings, and percentage of residents on total SB values, as affected by the Illumination × Sa presence interaction. Data are means of two sites and three replicates (*n* = 6). Within each row, values followed by the same letter are not significantly different at *p* ≤ 0.05, Tukey test. +Sa, presence of Sa adults; −Sa, absence of Sa adults.

	Full Sun		Canopy Shade	
SB Seedlings	+Sa	−Sa	+Sa	−Sa
**Number (n pot^−1^)**				
Residents	81.5 a	45.5 b	50.5 b	55.5 ab
*S. origanifolia*	5.5 a	2.5 a	6.0 a	4.5 a
Total	87.0 a	48.0 b	56.5 b	60.0 ab
Residents n° (%)	93.7 a	94.8 a	89.4 a	92.5 a
**Biomass (g dw pot^−1^)**				
Residents	8.0 a	9.6 a	3.8 a	9.9 a
*S. origanifolia*	1.8 b	6.9 a	1.9 b	8.7 a
Total	9.8 b	16.5 a	5.7 b	18.6 a
Residents dw (%)	81.8 a	58.3 b	66.9 ab	53.2 d
**N content (mg pot^−1^)**				
Residents	97.8 ab	105.3 ab	54.2 b	160.2 a
*S. origanifolia*	14.8 b	88.2 a	17.8 b	108.7 a
Total	112.6 bc	193.5 ab	72.0 c	268.9 a
Residents N (%)	86.9 a	54.4 b	75.2 a	59.6 b
**P content (mg pot^−1^)**				
Residents	25.3 a	27.9 a	11.6 b	32.1 a
*S. origanifolia*	3.8 b	16.3 a	3.9 b	18.2 a
Total	29.1 a	44.2 a	15.5 b	50.3 a
Residents P (%)	86.9 a	63.2 b	74.8 ab	63.9 b

**Table 5 plants-13-00226-t005:** Long-term-effect experiment: species or types that emerged from soil samples collected in April and June under established populations of *S. origanifolia* (Sa) and at a 2 m distance (Ctrl). A subset of soil collected in June was cold pre-treated (J 5 °C) for three months. x, species recorded in at least one tray.

		April		June		June 5 °C	
Species/Type	Family	Ctrl	Sa	Ctrl	Sa	Ctrl	Sa
*Dysphania botrys*	Amaranthaceae	x	-	-	-	-	x
*Periploca graeca*	Apocynaceae	x	-	-	-	-	-
unknown	Asteraceae	-	-	-	-	x	-
*Erigeron* sp.	Asteraceae	-	-	-	-	x	-
*Myosotis* sp. cfr	Boraginaceae	-	-	-	-	x	-
*Cardamine hirsuta*	Brassicaceae	-	-	-	-	x	x
*Cerastium glomeratum*	Caryophyllaceae	-	-	x	-	x	x
*Moheringia trinervia*	Caryophyllaceae	-	-	-	-	x	x
*Polycarpon tetraphyllum*	Caryophyllaceae	x	-	-	-	x	-
*Silene* sp.	Caryophyllaceae	-	-	x	-	x	-
*Stellaria media*	Caryophyllaceae	-	-	-	-	x	x
*Carex* sp. 1	Cyperaceae	x	-	x	-	x	-
*Carex* sp. 2	Cyperaceae	-	-	x	x	x	x
*Erica* sp.	Ericaceae	x	-	-	-	-	-
*Euphorbia peplus*	Euphorbiaceae	-	-	x	x	x	x
*Lotus* sp.	Fabaceae	-	-	-	x	-	-
*Trifolium* sp.	Fabaceae	-	-	x	-	-	-
*Juncus inflexus*	Juncaceae	-	-	-	-	x	x
*Oxalis corniculata*	Oxalidaceae	-	-	x	-	x	x
*Veronica arvensis*	Plantaginaceae	-	-	x	x	x	x
*Poa annua*	Poaceae	-	-	x	x	x	x
unknown	Poaceae	-	x	x	x	x	x
*Clematis vitalba*	Ranunculaceae	x	x	-	-	-	-
*Ranunculus* cfr *muricatus*	Ranunculaceae	-	-	-	-	x	x
*Solanum nigrum*	Solanaceae	-	-	-	-	x	-
unknown	Solanaceae	-	x	x	x	x	-
*Urtica dioica*	Urticaceae	-	-	-	-	-	x
Total n° of species		6	3	11	7	20	14

**Table 6 plants-13-00226-t006:** Short-term-effect experiment: main physical and chemical soil properties of the sites with low (LD) and high (HD) coverage of *S. origanifolia* in the San Bartolomeo Forest. Soil samples were collected on 5 March from the 0–10 cm profile.

Soil Properties	Unit	LD	HD
Sand (2 mm > Ø > 0.05 mm)	%	77.6	87.7
Silt (0.05 mm > Ø > 0.002 mm)	%	16.3	6.9
Clay (Ø < 0.002 mm)	%	6.1	5.4
Soil Organic C	%	13.9	9.3
pH	-	6.6	6.6
Total N	g kg^−1^	18.7	11.2
C/N	-	7.3	8.1
Cation Exchange Capacity	µS cm^−1^	349.5	266.0

**Table 7 plants-13-00226-t007:** Short-term-effect experiment: illuminance (lx) and % illuminance reduction in CS, recorded in the full sun (FS) and canopy shade (CS) illumination treatments at leaf onset (21 March) and at complete leaf unfolding (2 May) of the plane tree.

Day Time (h)	Illumination	21 March	2 May
09:00 a.m.	FS	12,372	42,540
	CS	5720	847
	CS reduction (%)	54	98
12:00 a.m.	FS	37,733	59,333
	CS	18,067	9000
	CS reduction (%)	52	85

## Data Availability

The data presented in this study are available on request from the corresponding author. The data are not publicly available as they are part of a wider unpublished data set included in a PhD project.

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
