# Peer review of "The Novel Invader Salpichroa origanifolia Modifies the Soil Seed Bank of a Mediterranean Mesophile Forest"

_plants, 2024, doi:10.3390/plants13020226_

Round 1

Reviewer 1 Report

Comments and Suggestions for Authors

Dear editor and authors,

I have read the manuscript “The novel invader Salpichroa origanifolia modifies the soil seed bank of a Mediterranean mesophile forest”. The authors surveyed the composition and structure of soil seed banks provide insight into the long-term implications of plant invasions on resident communities. Seed bank abundance and richness were also determined from, under and 2-m apart.  Manuscript is well written. The topic is interesting and new. I did have a few points that require clarification:

Lines 80-91: please move this part to Materials and Methods section.

Lines 125-389: why results is second part of manuscript section? After intoduction should be present Materials and Methods section. General note: all tables and figures numbers should be reorganized in accordance with the order of the manuscript sections.

Line 135: Reorganize number of table

Line 148: Reorganize number of table

Line 155 and 156 and so on in the following lines: Reorganize numbers of tables anf figures

Line 395: ‘Landolt’ please add reference

Lines 520-690: Materials and Methods section should be after Introduction section. General note: all tables and figures numbers should be reorganized in accordance with the order of the manuscript sections.

Line 525: Reorganize numbers of figure

Line 545: Change ‘the special protected Forest of 544 San Bartolomeo’ to ‘The special protected Forest of 544 San Bartolomeo’

Line 584: Reorganize number of table

Line 601: ‘combination Site x Sa’ not understandable

Line 605: Reorganize numbers of table

Line 609: Reorganize numbers of table

Line 645: ‘S. origanifolia’ please use the species name or abbreviation throughout the manuscript. Mixing these two forms causes confusion. It is general note.

Line 655: Change ‘L’ to ‘l’

Line 659: Reorganize number of figure

Line 665: Reorganize numer of figure

Line 676: Please use reference

Line 692-696: Please move hypothesis to end of introduction part. Then discuse about it it in discussion part. Conclusion is not the best place to hypothesize.

Reviewer 2 Report

Comments and Suggestions for Authors

The manuscript by Iduna Arduini and Viola Alessandrini present a new study which examines the novel invader Salpichroa origanifolia modifies the soil seed bank of a Mediterranean mesophile forest. The manuscript fit well with the standards of Plants. The text is well organized, easy to read and the topic is clearly developed.

Comments:

In the abstract and in the conclusion, it have to be indicated in which region the researches have been done.

On my opinion this article deserves to be published after minor revision.
